# New Advances in Second Language Acquisition Methodology in Higher Education

**Blanka Klimova** * and **Marcel Pikhart**

Department of Applied Linguistics, Faculty of Informatics and Management, University of Hradec Kralove, 500 03 Hradec Kralove, Czech Republic; marcel.pikhart@uhk.cz
* Correspondence: blanka.klimova@uhk.cz

**Abstract:** This article summarizes new advances, as described by current research, in the methodology of teaching Business English as a lingua franca (BELF) in the era of mobile learning and provides the reader with hands-on strategies that are useful for BELF classes and applicable in distance learning. The primary objectives of this literature review are to explore the fundamental approaches which should help practitioners in their course preparation, development, and teaching. The paper provides the reader with the most up-to-date strategies for teaching BELF and brings ideas on how to utilize these principles in a mobile learning (m-learning) environment. The methods include a literature review of available articles exploring the research topic, i.e., BELF and its pedagogy, which was performed by finding relevant studies in the Web of Science and Scopus databases. The results indicate that there are three fundamental approaches recommended by the current research on the teaching of BELF, namely task-based activities/case studies, exploitation of authentic materials, and blended learning implementation. In summary, the paper provides the readers with an update on current approaches for teaching BELF in higher education when utilizing modern tools for foreign language learning, such as m-learning, blended learning, and hybrid learning.

**Keywords:** BELF; applied linguistics; teaching strategies; teaching approaches; ESL; L2 acquisition; EFL

## 1. Introduction

English presently dominates the world of business as it is the language of communication in multinational companies around the globe. Most transactions are generally conducted in English [1–4], which compels individuals undertaking business activities to communicate in English when engaging in commercial or industrial ventures. Research reveals that 75% of employers prefer employees who can communicate in English as efficiently as possible [5]. The type of English language and skill required for business communication is referred to in modern linguistic theory as Business English (BE). Business English is defined as *the type of English used in business contexts, such as international trade, commerce, finance, insurance, banking, and many office settings. It entails expectations of clarity, particular vocabulary, and grammatical structures* [6]. Ellis and Johnson [7] extend this idea by emphasizing the fact that to develop an efficient BE course, key components, appropriate materials, and relevant tasks must be selected in order to develop the learner and attain the objectives.

Business English comprises several typical linguistic aspects, such as specialist vocabulary, that qualify as an independent research entity [8]. It is a branch of applied linguistics, i.e., an academic discipline covering various linguistic aspects of the language reality aimed at bringing pragmatic principles to be utilized in any language reality. In addition, BE is closely connected with Content and Language Integrated Learning (CLIL) in the way that it also focuses on content, communication, cognition, and culture [9]. It can be argued that culture is both a medium and an underlying principle of communication as it is stated by the sociocultural approach, and learning a foreign language is framed by this

cultural conditioning. The sociocultural approach provides us with an understanding of a language as a cluster of culturally and socially interdependent realities that are described and narrated by the given language.

One of the elements of applied linguistics is to investigate the new position of Business English in the current academic discourse. The most recent example of such a conversation would be English as a lingua franca in the global business world. Business English is acknowledged as Business English as a lingua franca (BELF) and International Business English. It commonly refers to the interactions between non-native speakers of English, whereas International Business English refers to the interactions between non-native and native speakers of English [10]. Despite the above terms, research shows [10] that 80% of all business communication in English is performed without the presence of native speakers.

Although BELF is, nowadays, taught within different Business English courses [11–14], there is still a continuing need to respond to the present requirements of the global market. In addition, further investigations into the literature revealed no significant research into BELF as a school subject, especially in post-communist countries [11,15]. Martins [16] highlighted BELF as a *neutral*, *practical*, and *culturally diverse* communication pattern. He advocated that this branch of English language is neutral because speakers use a language other than their native voice; practical because speakers aim at achieving their objectives efficiently and in the shortest possible time, and culturally diverse because speakers come from different cultural backgrounds.

Gajst [17] further added that BELF promotes simple and clear English, different from simplified English such as Globish, which includes neither idiomatic expressions nor complicated phraseology. All these characteristics should be reflected in researching, analyzing, optimizing, and teaching BELF. At present, BE or BELF is one of the most rapidly growing fields in the area of English language teaching and applied linguistics. It is, in fact, the leading branch of English for Specific Purposes (ESP) [18].

Most of the teaching strategies in BELF result from the teaching of ESP (TESP), which include the following: First, the satisfaction of students' immediate needs; second, the application of authentic materials; and finally, the issue of intercultural awareness. These are crucial aspects for the successful implementation of its methodology and must be taken into consideration when preparing BELF courses. Moreover, intercultural awareness is one of the most critical features of its methodology due to the trends in the global market. The importance of this feature has been highlighted earlier by relevant research [1,4].

BELF presents a challenge for researchers and teachers, particularly those involved in the teaching of English for General Purposes (EGP). Therefore, it is useful to clearly define the conditions for its teaching methodology as follows:

- Learners employ English to achieve their discipline-/job-specific needs;
- Learners are more autonomous and faster than other learners of English because BELF courses tend to be short;
- Learners have more significant opportunities to exploit modern information technologies because BELF courses are at the forefront of implementing these technologies in language teaching [19];
- Learners are exposed to a multitude of individuals with diverse cultural backgrounds when engaging with BELF speakers [20];
- Teachers work with authentic materials more than general ELT teachers since these materials demonstrate how real-world assignments are perceived and handled in their subject disciplines or their job.

The objectives of the paper are as follows. This paper aims at promoting novel ideas on improvising BELF instruction via practical and successful teaching methodologies, supported by current research on BELF methodology. Thus, the paper attempts to summarize the most useful pragmatic principles of BELF that could be employed when creating curricula for business, finance, ICT, and related areas in language education. In summary, this paper provides readers with an overview of a summary based on a literature review of

the potentially most efficient methods that can be utilized in the pedagogical approaches of BELF.

## 2. Methodology

This article reviewed the literature on related studies available in two databases, i.e., Web of Science and Scopus. The search collocations were as follows: *business English as a lingua franca* AND *methodology; business English* AND *methodology; business English as a lingua franca* AND *teaching.* Furthermore, the reference lists of the identified studies were investigated in order to not omit other vital studies on the research topic.

The following are the search results: Altogether, 1028 articles were detected in the Web of Science and Scopus databases. Fourteen articles were from the referenced articles and web-based studies. Most of the articles were located in Scopus (673), while 355 articles were sourced from the Web of Science. After removing duplicates and titles/abstracts unrelated to the research topic, 232 English-written studies remained. Of these, only 65 articles were relevant to the research topic. After excluding another 14 articles due to their irrelevant content and one descriptive study, 50 studies remained for the final investigation. The selection of these articles was done manually based on their clear relevance to the topic of BELF and FLL (foreign language learning). Some of the generated articles did not comply with the search criteria as many of them contained the key words indicating relevance but, in reality, they focused on a rather different topic loosely related to BELF. In addition, a Google search was conducted in order to detect unpublished (gray) literature. BK and MP individually performed an independent quality assessment of these 50 studies. They carefully read the articles to assess eligibility and to determine the importance of the paper for the researched topic. These essential quality criteria were selected using the Health Evidence Quality Assessment Tool for review articles.

Thus, 50 studies were investigated in full and they were considered against the following inclusion and exclusion criteria. The inclusion criteria were as follows: First, only studies involving business English as a lingua franca and its pedagogy were included. Second, the search was not limited to any time period. Third, both empirical and review studies were considered. The exclusion criteria were as follows: First, studies outside the scope of the research topic were excluded. Second, pure abstracts or descriptive studies, not involving any methodology of processing data, were excluded. Considering the above-described criteria, 28 studies were eventually included in the final analysis.

The major objectives of this literature research were as follows. First, to explore the fundamental approaches which should help BELF practitioners in their course preparation, development, and teaching; and second, to present a course design which reflects the primary methodological conditions and principles for teaching BELF.

## 3. Results

The findings indicate that the 28 studies on BELF methodologies have elaborated on several strategies, techniques, and approaches in the teaching of this new concept. The practitioners are confronted with massive productions of new textbooks on Business English or English as a lingua franca. Nevertheless, not many of these textbooks meet the specific needs of BELF students as the focus is on general business topics and skills to suit all business learners. Moreover, research on its methodology is limited; hence, this study was conducted as an attempt to deliberate on this important topic [21].

Based on the literature search, this paper highlights three approaches that are most recommended and utilized, namely task-based activities/case studies, exploitation of authentic materials, and implementation of blended learning, which should assist BELF practitioners in their course preparation, development, and instruction. Several other approaches include the integration of corpus linguistics, machine learning, and artificial intelligence into the learning process. These approaches, however, are more research-based and specifically related to the study of the language itself.

The recent literature and research summarize and recommend these three methodologies that have been tested and proved crucial in the BELF methodology. The results could be transferable into the everyday learning methodologies of BELF. The results are further elaborated in the following subsections that summarize the most useful methods applicable in its teaching process.

### 3.1. The Use of Case Studies

Case studies are advantageous and bring hands-on experience into the teaching methods of BELF. The focus is on interculturality, as these case studies refer to the global context. A case study is one of the examples of task-based activities that aim to solve real-world issues through communication. The task-based teaching methodology focuses on implementing real-life scenarios into the teaching process, thus making the learning process more connected to real global business contexts. It engages the students as they are required to solve business issues using the target language acquired in Business English classes and apply the knowledge of their subject areas [22]. The students are presented with the following stages of a case study: introduction, problem definition, problem solutions, presentation of the solutions, evaluation of the solutions, follow-up (optional), and feedback [23].

This approach presents several benefits for the teaching of BELF. First, it enhances team-working skills where learners develop cooperative learning in solving the problem; second, it fosters managerial skills where learners enhance their decision-making, problem-solving, and thinking skills; third, it strengthens learners' perception of their specialized subject knowledge by the case study issue; and finally, it exposes learners to varied accents, language structures, and collocations, e.g., while they are discussing the issue, reading it, or writing a report on it to complete the case study.

Furthermore, Chan [21] investigated task-based language learning in Business English contexts from learners' perspectives of task difficulty and their motivation to work on the given task. She revealed two crucial elements for successful implementation of task-based activities in language classroom learners' motives and task design. She also reported that the learners' history, including prior learning experiences and future career, had a direct impact on the level of task difficulty and motivation to perform the task.

It should also be noted that a case study approach has been criticized as lacking in sensitivity to the social and cultural dimensions of language learning because it requires consideration of these dimensions [24]. However, we claim that case studies are an ideal means to implement interculturality into the curriculum. Furthermore, with specific regard to teaching BELF, the naturalistic bias of the described approach has been deemed inefficient for teaching basic grammar and vocabulary for the beginners' level [25]. This problem of adequate student language proficiency and task complexity has been revealed by Guiyu and Yi [26]. The authors suggested that teachers optimize the cases and enhance mutual communication to achieve better teaching results. Moreover, cultural experience and also critical thinking experience could enhance learners' knowledge of BE/BELF.

Several BELF teachers may feel uncomfortable since they do not possess relevant content knowledge. Such a lack of experience would limit their performance [27], but this can be solved by collaborating with the corresponding subject teacher—team-teaching or collaborative team-teaching (co-teaching) are recognized forms of teacher collaboration which can be utilized for such purposes. Lately, team-teaching has become fashionable and encouraged in TESP or the teaching of BELF. Teachers have the options of engaging in three levels of collaborations, each with an increasing level of interaction [28]: collaboration, which includes the collection of information from the subject specialist department about curricula, assignments, and other information beneficial for BE course design; cooperation, which requires the BELF and the subject teachers to work together in setting up the BELF course and enhancing the subject-specialist course; and team-teaching. However, the collaboration usually reaches level one since subject teachers are less willing to cooperate with their BELF counterparts.

### 3.2. The Use of Authentic Materials

There is a high demand for the use of authentic materials in teaching BELF compared to the teaching of general English because learners are either professionals working in different businesses or people preparing for these professions [29]. Authentic materials are a relevant source that could be used in BELF task-based learning. They provide both teachers and students with articles related to real-life scenarios. Due to the nature of today's business, authentic materials also offer a platform for intercultural topics to be introduced in the course. Apart from that, these materials can be utilized as case studies which emphasize the importance of interculturality [29,30].

Case studies, as well as other materials in BELF, have always employed the use of authentic materials [28]. However, as research indicates [6,31], business English textbooks do not incorporate research findings from the field of business. As a result, there is a lack of authenticity in many of the textbook materials. Therefore, textbooks are not attractive enough for learners because they do not portray real-life situations. In this regard, Chan [31] suggested using transcripts of authentic workplace talk in teaching spoken business English to raise learners' awareness of features of spoken workplace discourse. In this way, learners become more aware of the relational side of workplace talk, as well as its associated discourse features, such as informal language, vague language, hedges, intensifiers, idioms, and pragmatic markers.

Another example of authentic materials used in BELF classes are videos of original materials [32]. Learners can listen, watch, and discuss current issues from the business world from those videos. Afterward, they carry out different tasks assigned by the teacher. These tasks correspond to real situations, which stimulate students to learn. Advertisements are another type of authentic material that can be utilized in the teaching of BELF. Lazovic [33] describes the use of advertisements in teaching BELF. She provides specific examples for learning and teaching vocabulary and grammar structures, improving oral and written skills, as well as intercultural awareness of her learners.

### 3.3. Blended Learning Implementation

Technologies in teaching BELF play a more crucial role than in any other branches of ELT since BELF situations are generally better resourced. On top of that, present students grow up surrounded by and exposed to technology. Hence, teachers need to be flexible and adaptable to students' learning preferences. Besides, it should be noted that students studying part-time engage in a lot of self-access language learning [34]. The blended learning approach is a suitable approach in addressing students' digital preferences and their part-time learning commitment.

In most cases, it is used as an additional supporting strategy to traditional, face-to-face teaching to enhance the BELF skills and language knowledge taught at school through the online component of the blended learning approach. Overall, students welcome this form of learning [35,36]. For instance, De Praetere [37] suggests that when teaching English as a foreign language, speaking and listening skills should be taught face-to-face while reading and writing can be performed online.

Further research [36] provides another example of a blended learning course of Business English aimed at both full-time and part-time students of Management of Tourism in the third year of their study at the Faculty of Informatics and Management (FIM) in Hradec Kralove, Czech Republic. No particular textbook was used, and all the materials in the online course were continuously modified. Students referred to the materials from this online course for their in-class and at-home studies. Teachers provided almost immediate feedback on the students' assignments or responded to their questions online. The blended learning approach offered students additional examples and exercises for practicing and revising the material taught at school, and students could access it anytime and anywhere.

Mobile learning, as a branch of e-learning [38], has become a new approach to teaching and learning [39] due to the widespread use of smartphones among the younger generation. Thus, blended learning appears to be a combination of traditional, face-to-face formal

instruction with the use of mobile applications in informal settings outside the classroom. However, in comparison, with e-learning, learning phases are shorter, but they are more frequent [40,41]. Blended learning enables more frequent learning that has a positive impact on the retention of new knowledge. Despite this, a blended learning approach may be time-demanding due to its preparation and management [42]. It may also demotivate students who are less motivated and lack self-discipline to work independently at their own pace.

*3.4. Course Design Suggestions*

The following ideas summarize the most important strategies which should be implemented into the BELF methodology to create an impactful learning environment. The findings of the research support these recommendations. In designing a course, the most important aspect is to analyze students' needs before the commencement of the course to discover their strengths and weaknesses, as well as to be able to set course objectives and the key topics. This might take different forms, which can be as follows: First, performing a diagnostic test (DIALANG) to identify students' strengths and weaknesses in the area of language proficiency and set relevant students' levels of English [43]; second, conducting a questionnaire survey [11]; third, performing a strengths, weaknesses, opportunities, and threats (SWOT) analysis; fourth, engaging in observations [11]; fifth, analyzing language corpora [11]; and finally, conducting informal consultations with other language teachers and subject specialists.

After specifying the course objectives, the syllabus can be developed. In the case of BELF, it can either be a task-based syllabus or a content-based syllabus, or a mixture of both. Moreover, Chan [21] emphasized that the syllabus should be flexible to enable teachers to make modifications in response to learner factors. These factors involve learners' histories and the motives which they bring to their learning.

The next important step involves the development of relevant materials for each syllabus task or topic. The current research [44,45] proposes the following structure for the development of any topic-based study materials that would be a two-page document consisting of the following items:

1. Topic (the central theme of the lesson described in one sentence);
2. Learning objective (a short statement motivating the participants to study the particular lesson);
3. Preconditions (previous knowledge required to master the lesson);
4. Skills (a description of the knowledge/skills to be gained in the specific lesson);
5. Explanation of the basic concept and ideas of the teaching matter discussed in the lesson (in the form of text and questions);
6. Conclusion with self-tests, tasks, quizzes (with keys), or an assignment;
7. Bibliographical sources or links to them.

The following step is to deliver this material in a classroom. As indicated above, there are different approaches to BELF, and it depends on the students' needs, situation, and context. It is the task of the teacher to select the most suitable approach according to the learning situation and to ensure that students acquire the learning material and practice it. The course design should also be based on current research outcomes and should ideally be supported by several empirical studies and their methodology [46,47].

As for the assessment, students should be tested continuously. Therefore, formative assessment should be implemented since continuous testing stimulates long-term retention of language knowledge and skills. Furthermore, it enhances students' learning performance because students can observe their ongoing learning progress and become actively involved in their learning. This assessment is also very beneficial for BELF teachers because it provides them with feedback on their work, as well as on students' learning performance. More specifically, it gives them information about students' strengths and weaknesses as far as BELF learning is concerned. This feedback can be carried out in various ways; by testing students' knowledge, questionnaires, observing students' performance in class or

the online course, or directly through students' reflections on the course in the form of a written essay.

## 4. Discussion

The current pedagogy and learning psychology reflect the need to implement blended learning methods into the instructional process. Adoption of the new course design [48] ensures that institutions keep pace with the latest trends and development of human–computer interaction and all the consequences which it bears [49]. Online learning approaches are necessary for the new generation of technologically savvy learners. The current research also stresses the importance of this implementation for the new generation of learners, the so-called Generation Z, who have specific needs and approaches to communication and learning [50]. These new approaches will create an environment that will be learner-friendly and also stimulating for this new generation who disfavor the traditional and conservative teaching methods.

Lichterfeld [15] reports that in teaching BELF, there is a general shift from a native speaker model to intelligibility. In this sense, pronunciation plays an important role together with excellent listening skills, familiarity with different accents, and cultural awareness. As she puts it: *BELF is not culturally neutral or "conflict-free". Our learners have to become aware of the importance of intelligibility, credibility, adaptability, accent, and identity.* As far as cultural awareness is concerned, Chong [51] proposed the awareness, do not judge, analyze, persuade yourself, and try (ADAPT) model.

The most crucial aspect of this concept as a subject is the pragmatic essence on which it is based. The practical purpose is the underlying theme, which must be considered and respected when creating the curricula of various academic programs, including BELF. There could be further research into the efficiency of these approaches. It is crucial to focus on the level of improvement of the teaching process after implementing these methodologies and not just blindly accept that blended learning and e-Learning will overcome the other methods.

The most significant limitation of this research was the risk of bias in individual studies. However, this limitation was not substantial for the findings and did not present any severe distortion of the information yielded. Another limitation was the risk of bias that might affect the cumulative evidence, such as selective reporting or cross-referencing within studies. Again, this did not significantly influence the results of this research.

BELF has recently become a crucial subject for many academic disciplines including business studies, finance, insurance, and international trade. However, there are certain areas where the urgent need to implement it into curricula is more than evident, such as ICT. This article is an attempt to highlight the importance of such an implementation to benefit from enhanced global cooperation supported by improved communication patterns and approaches.

Thus, if BELF teachers take on the challenge to respond to students' immediate and specific needs, continuously modify and revamp their personalized materials, and carry out continuous assessments, their teaching and students' learning might be effective and successful, especially if they feel motivated [7]. Due to the current massive utilization of information technologies in learning, we are facing new challenges, and education has recently become dramatically different from a few years ago in favor of maximizing distance learning opportunities. Therefore, reconsidering the possibilities of e-Learning utilization is crucial for future educational sustainability.

**Author Contributions:** Conceptualization, B.K. and M.P.; methodology, B.K. and M.P.; validation, B.K. and M.P.; formal analysis, B.K. and M.P.; investigation, B.K. and M.P.; resources, B.K. and M.P.; data curation, B.K. and M.P.; writing—original draft preparation, B.K. and M.P.; writing—review and editing, B.K. and M.P. All authors have read and agreed to the published version of the manuscript.

**Funding:** This research received no external funding.

**Institutional Review Board Statement:** Not applicable.

**Informed Consent Statement:** Not applicable.

**Data Availability Statement:** Not applicable.

**Acknowledgments:** This paper was supported by the research project SPEV 2021, run at the Faculty of Informatics and Management, University of Hradec Kralove, Czech Republic. The authors thank Aleš Berger for his help with data collection.

**Conflicts of Interest:** The authors declare no conflict of interest.

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
