# Peer review of "New Advances in Second Language Acquisition Methodology in Higher Education"

_education, doi:10.3390/educsci11030128_

Round 1

Reviewer 1 Report

General feedback

There are two major issues with this submission, one of which is related to the framing and the other is related to methods.  The framing issue relates to the authors' model of language learning.  Throughout, the line between content knowledge and language knowledge is blurred, often substituting learning about the culture or solving problems for learning the language.  It is possible to argue that culture is a medium for acquisition (e.g., sociocultural approach), but this argument is not made.  Instead, there is an implicit assumption that by practicing solving problems, the learners will become better with Business English/BELF, and the suggestions to other instructors then revolve around better ways to grapple with technology and problem-solving.  Although these are facilitative, they are not language unto itself, even Business English/BELF.  References to the acquisition of the language (e.g., phonetics, grammar) are made as supplementary commentary towards the end of these sections, suggesting that the focus of the instruction is elsewhere.  Again, if the framework were explicitly laid out as something like sociocultural approach, the submission as a whole would be more convincing.

The methodological issue is in the selection of the review articles and how they inform the results.  Although I include specific critiques of the filter process below, the criteria for selection is vague and no examples are given.  A particular criteria related to relevancy appears twice with two different stages of filtering, and they aren't transparently different.  Similarly, the methods lists one final number of studies, and the results lists another.  The methods of this paper need significant revision.  For example, if the Health Evidence Quality Assessment Tool was instrumental, this should frame the whole section and specific criteria from the tool should be used.  The HEQAT's relationship to what's being done is somewhat opaque.  As to how they inform the results, specific examples would be helpful.

Specific Feedback

  1. Line 87 – Claims that BELF is “a new systematic approach in the study of the pragmatic aspects of the language”  What is this claim made upon?  New in what scope?  What pragmatic aspects?
  2. Motivation/relevance of social constructivism theory (lines 102-5) unclear; perhaps it’s true that this theory emphasizes this, but what does it have to do with this paper?
  3. Article selection unclear in lines 114-6; 65 were relevant but of those 14 removed due to irrelevant content; how are these different?
  4. Article selection in lines 122-129 unclear; unclear how these criteria are distinct.  That is, how are studies that “involv(e) business English as a lingua franca and its pedagogy” different from those that remain when studies were removed for being “outside of the research topic.”  Also, how are the studies removed in these lines for being descriptive different from the one removed in line 116?
  5. The methods indicate that only 8 remained but 50 are mentioned at the beginning of the results.  Which set are those that are used in the discussion of 3.1-3.3?  Which of the references at the end of this article are in these sets?  Is it possible for the author(s) to include a number range of those references at the point in which they’re relevant?
  6. Claim on line 149 that these are “efficient” methodologies.  No evidence presented comparing efficiency.
  7. The argument made for case studies is not one for the acquisition of BE/BELF but for the acquisition of adjacent skills; the author(s) themself/themselves acknowledge that the linguistic dimensions are not served with this approach (lines 183-6).  It would be possible to argue that the cultural experiences (lines 180-182) or critical thinking skills would enhance learners’ knowledge of BE/BELF, but the author(s) does/do not make this argument.
  8. Sections 3.2 and 3.3 still not clear as to the English piece, framework of acquisition implicit
  9. Sections 3.4 trying to make connections without making it clear what they're connecting. In particular, they say "as we said above" without me being able to find out where above
  10. Pronunciation appears in discussion as afterthought

Author Response

Dear Reviewer,

Please see the attachment about our modifications and changes.

Regards,

Authors

Reviewer 2 Report

  • Line 8: Should be a literature review.
  • Line 10: What is an m-learning?
  • Improve the abstract. Should be clear and give some background, your objectives, and major conclusions.
  • Line 25: "More or less" does not sound a good choice for scientific writing.
  • Line 92: your objectives are not clear. Restate your objective in a clear way.
  • Line 130: It should go to line 92. Why do you repeat objectives?
  • Your conclusions are no real conclusions. It looks more like discussion

Author Response

(The authors gave the same response as above.)

Round 2

Reviewer 1 Report

Thanks to the author(s) for their work.  The changes in this paper reflect previous commentary.  The addition of the framework and clarifications on methods are sufficient for publication.

Reviewer 2 Report

Thank you for addressing my comments.